# Evaluation of Lower Leg Arteries and Fibular Perforators before Microsurgical Fibular Transfer Using Noncontrast-Enhanced Quiescent-Interval Slice-Selective (QISS) Magnetic Resonance Angiography

**DOI:** 10.3390/jcm12041634

**Published:** 2023-02-18

**Authors:** Annett Lebenatus, Karolin Tesch, Wiebke Rudolph, Hendrik Naujokat, Ioannis Koktzoglou, Robert R. Edelman, Joachim Graessner, Olav Jansen, Mona Salehi Ravesh

**Affiliations:** 1Department of Radiology and Neuroradiology, University Hospital Schleswig-Holstein, Campus Kiel, 24105 Kiel, Germany; 2Department of Oral and Maxillofacial Surgery, University Hospital Schleswig-Holstein, Campus Kiel, 24105 Kiel, Germany; 3Department of Radiology, NorthShore University HealthSystem, Evanston, IL 60201, USA; 4Pritzker School of Medicine, University of Chicago, Chicago, IL 60637, USA; 5Feinberg School of Medicine, Northwestern University, Chicago, IL 60611, USA; 6Siemens Healthcare GmbH, 20910 Hamburg, Germany

**Keywords:** anatomic branching pattern of lower leg arteries, fibular perforators, microsurgical fibular transfer, non-contrast-enhanced quiescent-interval slice-selective (QISS), patients with oral and maxillofacial tumors

## Abstract

(1) Background: Preoperative imaging of the lower leg arteries is essential for planning fibular grafting. The aim of this study was to evaluate the feasibility and clinical value of non-contrast-enhanced (CE) Quiescent-Interval Slice-Selective (QISS)-magnetic resonance angiography (MRA) for reliably visualizing the anatomy and patency of the lower leg arteries and for preoperatively determining the presence, number, and location of fibular perforators. (2) Methods: The anatomy and stenoses of the lower leg arteries and the presence, number, and location of fibular perforators were determined in fifty patients with oral and maxillofacial tumors. Postoperative outcomes of patients after fibula grafting were correlated with preoperative imaging, demographic, and clinical parameters. (3) Results: A regular three-vessel supply was present in 87% of the 100 legs. QISS-MRA was able to accurately assign the branching pattern in patients with aberrant anatomy. Fibular perforators were found in 87% of legs. More than 94% of the lower leg arteries had no relevant stenoses. Fibular grafting was performed in 50% of patients with a 92% success rate. (4) Conclusions: QISS-MRA has the potential to be used as a preoperative non-CE MRA technique for the diagnosis and detection of anatomic variants of lower leg arteries and their pathologies, as well as for the assessment of fibular perforators.

## 1. Introduction

In 1989, the free fibular flap (FFF) described by Taylor in 1975 was first used by Hidalgo for mandibular reconstruction [1,2]. These flaps can be harvested as vascularized bone flaps or osseoseptocutaneous flaps to cover defects of the orofacial skeleton after oncological resection, trauma, infection, or osteonecrosis [1,3]. In mandibular reconstruction, the FFF represents the option of choice for many surgeons because, owing to the amount of bone, even multi-segment defects can be treated; it also provides a sufficient vascular pedicle, prosthetic rehabilitation is possible with dental implants that give stable long-term results, and it can be harvested in a two-team approach [4,5]. The skin island of the fibular flap is usually perfused by septocutaneous perforators, whereas the fibula itself is supplied by segmental periosteal branches, all of which are fed by the fibular artery, which is harvested with the free flap for revascularization [6].

Perfusion of the foot is primarily secured by the anterior and posterior tibial arteries. In the case of either congenital anomalies or atherosclerotic disease of the tibial arteries, the fibular artery may become the major feeding artery.

However, the fibular artery might also be congenitally absent or abnormal. Under any of these anatomical conditions, removal of the fibular vessels with the FFF would jeopardize the donor leg, the fibular flap, or both [7,8]. The most feared complication of a fibular graft is ischemia of the ipsilateral foot after removal of the fibular artery. For the viability and thus success of osseoseptocutaneous flaps, it is important to select a graft segment that contains one or more septocutaneous perforators to ensure that the grafted skin island is adequately perfused [6,9]. In many cases, the fibular perforators are located over the third quarter of the fibula; however, the count and location can vary and are not reliable [10].

For the aforementioned reasons, the standard of care involves preoperatively evaluating the arteries of the lower leg. In clinical routine, contrast-enhanced computed tomographic angiography (CE-CTA) or magnetic resonance angiography (CE-MRA) is usually performed [8,11].

Oncologic patients undergo multiple preoperative and subsequent imaging examinations due to their diseases. Therefore, it is desirable to perform CE-MRA examinations instead of CE-CTA examinations to avoid the adverse effects of radiation exposure and iodine-containing contrast agents [9,12].

However, it has been known since 2013 that gadolinium-based contrast agents (GBCAs) used for MRA examinations can be deposited in the brain as well as in other organs [13,14]. GBCAs should also be avoided in patients with impaired renal function; therefore, MR techniques that do not use any type of exogenous contrast agent are of considerable interest.

In 2010, Edelman et al. presented the 2D Cartesian electrocardiogram (ECG)-triggered Quiescent-Interval Slice-Selective (QISS) pulse sequence as a non-invasive MRA technique for the visualization of lower leg arteries. Since then, studies have shown the reliability of this technique for the non-CE visualization of leg arteries compared to CE-MRA as well as CE-CTA and intra-arterial catheterization angiography [15,16,17].

The purpose of this study was to investigate the feasibility and clinical value of QISS-MRA for reliably visualizing anatomical branching patterns of the lower leg arteries and the fibular perforators. The postoperative outcome of fibular grafting was compared with the preoperative assessment of lower leg arteries using QISS-MRA.

## 2. Materials and Methods

### 2.1. Patients

From November 2018 to October 2021, a total of 50 patients were consecutively referred from the Department of Oral- and Maxillofacial Surgery to our Radiology Department. Patients suffered from different types of oral cancer, odontogenic tumors, and bony disease that might result in mandibular segmental resection; therefore, the iliac and leg arteries were evaluated preoperatively for potential fibular grafting. The general exclusion criteria for MRI safety were large-sized ferromagnetic materials or non-MRI compatible implants in the body and further contraindications to MRI. Relevant demographics (age, weight, body mass index at examination, and gender) and clinical data of the study population are included in Table 1.

### 2.2. MRA Imaging

Imaging was performed in transverse slice orientations using QISS-MRA with the parameters shown in Appendix A on a 1.5T MRI system (Magnetom Aera, XQ gradients, syngo MR VE11C software, Siemens Healthcare, Erlangen, Germany) with a maximum gradient of 45 mT/m and a maximum slew rate of 200 T/m/s. The MRI signal was received using a 36-element dedicated peripheral angiographic coil from the same manufacturer. Depending on the length of the patients’ lower extremities, a total of 8 to 10 measurement slabs with 60 slices each were acquired.

### 2.3. Image Analysis

The datasets of these 50 patients were evaluated mainly independently in separate sessions by two radiologists (A.L. and K.T.), both with 7 years of experience in vascular imaging. The anatomical branching pattern of the lower leg, the length of the tibiofibular trunk, and the length of the fibula were determined by consensus of the two radiologists. MRA datasets were analyzed from both a technical and a clinical perspective. The two radiologists (A.L. and K.T.) rated the technical quality of the MRA images of the lower legs using a scoring scale with respect to various imaging artifacts (Figure 1):
Grade 0: Absence of imaging artifactsGrade 1: Presence of
(a)Venous contamination,(b)Signal dropout due to endoprostheses or other ferromagnetic implants,(c)Stair-step artifacts,(d)Motion artifacts,(e)Different signal intensities within a measurement slab.

The apparent signal-to-noise ratio (SNR) and contrast-to-noise ratio (CNR) were computed as described in the Appendix A to evaluate the quality of lower leg artery visualization by using QISS-MRA. Here, the anterior tibial artery is abbreviated as AT, posterior tibial artery as PT, and fibular artery as FA.

SNR and CNR were determined using the extracted mean value and standard deviation (sd) of the signal intensities (SI) from these ROIs as follows [18]:$$SNR=0.655\cdot \frac{mean\;(SI_{\mathrm{Vessels}})}{sd\;(SI_{\mathrm{Air}})},\;CNR=0.655\cdot \frac{mean\;(SI_{\mathrm{Vessel}}-SI_{\mathrm{Muscle}})}{sd\;(SI_{\mathrm{Air}})}$$

Clinical evaluation was divided into the following five sections:

(1) Classification of the anatomical branching pattern of the lower leg arteries according to Kim et al. [19] as follows:Regular pattern with three supplying arteries (Type I-A),Irregular pattern withTrifurcation of the three lower leg arteries (Type I-B),First exit of PT below the knee joint, then the joint exit of FA and AT via a common trunk (Type I-C),AT arises above the knee joint (Type II-A),As I-C, but PT exits at the level of the knee joint (Type II-B),The FA arises above the knee joint (Type II-C),Hypoplastic PT (Type III-A),Hypoplastic AT (Type III-B),Hypoplastic PT and AT (Type III-C), also called “PAM” (peroneal magnus artery).

(2) The length of the fibula was measured in millimeters over its entire length from the fibular head to the lateral malleolus.

(3) The length of the tibiofibular trunk was measured in millimeters from the exit of the anterior tibial artery from the popliteal artery to its branch into the posterior tibial and fibular artery.

(4) The preoperative status of all three lower leg arteries (ATs, PTs, and FAs) was assessed by grading their visibility and scoring their patency.

Visualization of ATs, PTs, and FAs was evaluated by assessing continuity, visibility, and edge sharpness using a 1 to 4 grading scale for image quality (Figure 2):Grade 1: Nondiagnostic, barely visible lumen rendering the segment.Grade 2: Fair, ill-defined vessel borders with suboptimal image quality for diagnosis.Grade 3: Good, minor inhomogeneities not influencing vessel delineation.Grade 4: Excellent, sharply defined arterial borders with excellent image quality for highly confident diagnosis.

The presence or absence of stenosis in the respective lower leg artery was assessed by evaluating source images in the transverse plane and 3D reconstructions (Maximum Intensity Projection, MIP). Stenosis was rated using a scoring scale of 0 to 2:Score 0: no stenosis.Score 1: presence of a hemodynamically relevant stenosis in the

Score 1A: proximal third, or

Score 1B: middle third, or

Score 1C: distal third of a lower leg artery.

Score 1AB: proximal and middle third of a lower leg artery.

Score 1ABC: proximal, middle, and distal third of a lower leg artery.

The degree of stenosis could not be determined if an artery was missing due to anatomic variance or previous surgery.

(5) The presence, number, and location of fibular perforators were determined. Localization was indicated by the distance in mm between the exit of the perforator from the fibular artery and the lateral malleolus.

In total, 100 lower legs were included in our study, and each radiologist conducted a total of 400 analyses for evaluating imaging artifacts, grading and scoring of lower leg arteries, and determining fibular perforators. A total of 300 additional analyses were performed by consensus of the two radiologists to determine the anatomical branching pattern of the lower leg, the length of the tibiofibular trunk, and the length of the fibula. Image analysis was performed on a workstation (IMPAX EE, Agfa HealthCare GmbH, Bonn, Germany).

### 2.4. Statistical Analysis

Normality (Gaussian distribution) was tested for all variables using the Shapiro–Wilk test. The median and range of variables were reported as summary statistics for all variables due to the severe skewness of the respective distributions. The *p*-values for the comparison of two groups were obtained by using a Wilcoxon signed-rank test. A *p*-value < 0.05 was considered statistically significant; *p*-values were reported with a precision of 10^−5^. The surgical outcomes of 25 patients with fibular grafting were correlated with demographic and clinical parameters using Fisher’s exact test or Spearman’s correlation test [20]. The strength of the Fisher’s exact correlation was defined with a Phi-value [21]: If the variables are not associated, then the Phi coefficient should be 0; perfect positive (negative) association yields a Phi coefficient of 1 (−1).

The strength of the Spearman’s correlation was defined as perfect (r = 1.0), very strong (1.0 > r ≥ 0.80), moderate (0.80 > r ≥ 0.60), fair (0.60 > r ≥ 0.30), and poor (r < 0.30) [22].

Interobserver agreement on the scoring of an image was assessed by quadratic weighted concordance [23]. Concordance was calculated with the rater’s package (CRAN: raters). Agreement was interpreted as follows: <0.00, poor agreement; 0.00–0.20, slight agreement; 0.21–0.40, fair agreement; 0.41–0.60, moderate agreement; 0.61–0.80, substantial agreement; 0.81–1.00, or almost perfect agreement [24]. R software (version 3.5.1, R Foundation for Statistical Computing, Vienna, Austria) was used to statistically analyze our data and to generate the diagrams.

## 3. Results

### 3.1. Demographic and Clinical Characteristics of the Study Population

QISS-MRA was used to preoperatively evaluate lower leg arteries before microsurgical fibular grafting in 50 patients. All datasets were included in the present study. Relevant data for the study population are summarized in Table 1; 21 (42.0%) patients were female, and 26 (52.0%) patients were 65 years or older.

### 3.2. SNR and CNR

The determined median SNR and CNR values of all patients are summarized in Table 2. There were no differences between the SNR or CNR of the left and right sides of the popliteal artery, AT, FA, and PT. The median SNR values of aortic bifurcation and all aforementioned arteries were in the range of 290.7 [207.4, 404.3], and the CNR of the lower leg arteries was in the range of 260.4 [184.7, 375.6].

### 3.3. Anatomical Data

The median lengths of right and left fibulae were similar (380.0 [360.0, 405.0] vs. 381.0 [361.2, 402.8] mm, *p* = 0.82).

Detailed anatomic analysis of QISS-MRA source data of 100 legs according to the branching classification described by Kim et al. [19] revealed that in 87 cases (87.0%) a regular three-vessel supply was present (84.0% (42/50) in right legs and 90.0% (45/50) in left legs), in 5 cases (5.0%), a two-vessel supply (4.0% (2/50) in right legs, 6.0% (3/50) in left legs), and in no cases, a one-vessel supply. In eight legs, an irregular three-vessel supply was observed that was classified as type IC (two legs) or IIB (six legs). Four of five two-vessel supplies had a hypoplastic PT, and one leg a hypoplastic AT. Types I-B, II-A, II-C, and III-C were not identified in our patient population. Examples of anatomical vessel variations are demonstrated in Figure 3.

A regular three-vessel supply was observed in both legs of 16 female and 24 male patients from various age ranges.

There were no significant differences between the median lengths of the right and left tibial-fibular trunk (TFT) (36.0 [27.0, 44.0] vs. 30.5 [24.3, 43.3] mm, *p* = 0.34). In our patients, the range of TFT length was 1–137 mm. In 13 cases, the TFT was longer than 50 mm. An aberrant vascular supply led to missing values of TFT in 13 legs.

Fibular perforators were found in 87% of 100 legs (46 right legs and 41 left legs). Image examples of fibular perforators are presented in Figure 4.

The median number of perforators varied within patients (2 [1, 3]) and also between the right and left legs (2 [1, 3] vs. 3 [1, 3], *p* = 0.24). There were no significant differences between the two observers concerning the determination of perforators (Table 3).

The normalized perforator location in relation to the fibular length of individual patients and the number of found perforators in all patients are presented in Figure 5.

Most perforator vessels were found at the level of the second third of the fibular bone. In the 13% of legs that did not have perforators, fibular grafting was still conducted in seven patients without any subsequent complications such as graft loss.

### 3.4. Clinical Image Quality

Image quality was graded as “excellent” by both radiologists in 83.0% (166/200) of FAs, in 82.5% (165/200) of ATs, and in 79.5% (159/200) of PTs. In 16.0% (32/200) of FAs, 15.5% (31/200) of ATs, and 15.0% (30/200) of PTs, the image quality was rated as “good”. There were 0.5% (1/200) of PTs with a “fair”, 0.5% (1/200) of ATs, and 2.0% (4/200) of PTs with a “non-diagnostic” image quality (Table 3). Overall, image quality could not be evaluated in 5.5% (11/200) of vessels because the corresponding vessel was not present due to anatomic abnormalities or previous surgery.

There were no stenoses in 96.0% (192/200) of cases in FAs, 95.5% (191/200) of cases in ATs, and 92.5% (185/200) of cases in PTs (Table 4). In 2.0% (4/200) of cases, proximal stenosis was found in the FA, in 0.5% (1/200) of cases in the AT, and in 0.5% (1/200) of cases in the PT. Overall, 0.5% (1/200) of stenosis cases were observed in the second third in the FA, AT, and PT. Distal stenoses were found in 0.5% (1/200) of cases in the FA, 1.5% (3/200) of cases in the AT, and 1.0% (2/200) of cases in the PT.

The AT was stenosed in 0.5% (1/200) of cases and the PT in 1.0% (2/200) of cases along the entire vessel length. Combined stenoses in the proximal and middle thirds of PT were found in 1.0% (2/200) of cases. Overall, stenoses could not be evaluated in 5.5% (11/200) of vessels. Figure 6 shows an example of a long-distance stenosis of the posterior tibial artery.

### 3.5. Imaging Artifacts

Both radiologists evaluated the imaging artifacts in the lower legs independently and in separate sessions. The results are summarized in Table 4. Venous contamination was observed in 2.0% of 200 legs. Of these, only a very small percentage (2.0%) showed artifacts due to metallic implants. However, these artifacts were found in one lower leg of each of two patients. Stair-step artifacts were present in the majority of arteries (71.5%) but did not affect the clinical evaluation of the lower leg vessels. Motion artifacts (7.5% of cases) and different signal intensities within a measurement slab (8.0% of cases) were rarely observed.

### 3.6. Interobserver Agreement

Interobserver agreement for grading the image quality of lower leg arteries, for scoring the stenosis of these arteries, and for determining the amount and position of fibular perforators ranged from 84 to 100% with the exception of stair-step artifacts (Table 5).

### 3.7. Surgical Outcome

A total of 46 patients received surgical therapy, and 4 patients (8%) palliative treatment without surgery. In 30 cases, ablative surgery resulted in mandibular segmental defect, and reconstruction was performed using an FFF in 25 patients (50%), an iliac crest graft in 1 patient, a scapular transplant in 1 patient, and alloplastic reconstruction in 3 patients. The decision for the iliac crest graft was based on the geometry of the defect of the mandible, the scapula-flap, and the reconstruction plates on pathological vascular findings of the leg as well as reduced general health. In 16 patients (32%), segmental mandibular resection was not performed during surgical therapy. Therefore, no bone transplant was needed.

Evaluation of the FFF showed a postoperative complication rate of 36% (9/25), including insufficient perfusion of the graft, intra- and extraoral dehiscence, (partial) necrosis of the skin island, and delayed wound healing at the donor site. In seven cases, the FFF could be preserved as a result of surgical revision and prolonged wound care, which corresponded to a success rate of 92% (23/25). In two cases, the FFF had to be explanted. In the nine cases in which complications developed, no stenoses were found on preoperative MRI. All nine complication cases had fibular perforators.

Neither clinical parameters, such as diabetes, hypercholesterolemia, smoking, or arterial hypertension, nor the presence or the number of fibular artery perforators correlated with the surgical complications that developed (Appendix A).

## 4. Discussion

The FFF can currently be regarded as the “clinical standard technique” for mandibular reconstruction, and due to several modifications in harvesting and shaping, it has become the “workhorse“ for multiple indications [25]. Several examples of invasive, minimally invasive, and noninvasive imaging of the lower leg arteries prior to planned fibular grafting can be found in the literature [6,26].

The previous standard of invasive catheter angiography of the arteries of the lower leg has been replaced over the years by CE-CTA and CE-MRA due to its invasiveness and possibly severe procedural complications [27].

Especially in cancer patients, frequent and repeated administration of contrast agents is potentially harmful to the kidneys. Therefore, non-contrast imaging of the lower leg arteries would be desirable.

Ultrasound-guided imaging of the lower leg arteries is a noninvasive, inexpensive, and widely available method, but it is time-consuming and has some technical limitations such as a small field of view, low penetration depth, and strong operator dependence [28].

In 2010, Edelman et al. demonstrated that QISS-MRA provides imaging of the leg arteries that is equivalent to that of contrast-enhanced MRA [15]. To date, the QISS-MRA technique has not been evaluated for assessing vascular status before planned fibular grafting.

In our retrospective study, we investigated the feasibility and clinical value of QISS-MRA as a preoperative imaging technique in 50 patients with oral and maxillofacial tumors who were scheduled for fibular transplantation. The QISS-MRA datasets were evaluated from the clinical and technical perspective concerning image quality and vessel visualization. The focus of our study was to assess the classification of the anatomic branching pattern, evaluate the presence of stenosis, and visualize the fibular perforators of the lower leg arteries. These results were compared with the postoperative outcome of these patients.

Our study showed that (1) QISS-MRA provided high SNR and CNR, resulting in an excellent visualization of lower leg arteries without administering GBCAs. (2) QISS-MRA provided reliable arterial image contrast due to the suppression of venous signal intensity in almost all cases. (3) The absence of interfering imaging artifacts helped accurately classify anatomic branching patterns of the lower leg arteries. (4) For the first time, fibular perforators were depicted using a non-CE MRA technique.

Therefore, QISS-MRA has the potential to become the standard technique for evaluating lower extremity vascularity in patients with oral and maxillofacial tumors in whom fibular grafting is planned.

### 4.1. Technical Aspects

The prevalence of five distinct image artifacts during QISS-MRA was analyzed for lower leg arteries. Venous contamination was found in only one patient, presumably due to an insufficient ECG lead and resulting time delay in suppressing the venous signal. The absence of venous contamination in most of the datasets we obtained suggests that QISS-MRA should be preferred to CE-CTA and CE-MRA for imaging of lower legs arteries. In a total of two patients, we observed signal dropout in one lower leg artery, both due to a metal implant. To enable evaluation of vessels in close proximity to the metal implants, it could be helpful to implement a broadband excitation pulse or use a spoiled gradient-echo readout in the pulse sequence [29]. The occurrence of stair-step artifacts in approximately 70.0% of the lower leg arteries was due to geometric inconsistencies between adjacent slices in the reconstruction algorithm. However, these artifacts did not affect the diagnostic quality of the lower leg angiograms. Future work could address the improvement of 2D reconstruction algorithms or the investigation of 3D QISS-MRA and balanced T1 relaxation-weighted steady-state MRA methods to reduce or eliminate the stair-step artifacts [30,31]. In a few cases, motion artifacts were observed that could be eliminated by appropriate padding of the lower legs without compressing the vessels. Signal intensity changes within a measurement slab due to insufficient local magnetic field homogeneity were present in about 8.0% of arteries. The observed changes in the signal intensity along the lower leg arteries within a slab can be caused by the different shim status of the individual vessels. This artifact could be omitted by employing higher-order shimming for more accurate magnetic field correction.

When we evaluated the angiograms, we found that the sensitivity of the 36-channel coil we used decreased on the outer sides of the legs, deteriorating the image quality compared with the inner sides of the legs, especially in overweight patients with increasing leg circumference. The rigid central segment of this coil is placed between the patient’s legs and is found to be bothersome, especially by male patients. It would therefore be desirable to design a flexible coil in the form of trousers that would optimally cover different patients’ legs. With coil elements that can be connected with Velcro, it would be possible to adapt the width and length to different patients. A similar approach is applied for visualizing hand arteries using a glove coil [32].

### 4.2. Clinical Aspects

Owing to the predominantly good to excellent image quality (grading ≥ 3), the anatomical branching pattern could be classified and the degree of stenosis and fibular perforators assessed on the bilateral lower leg.

Precise classification of the anatomic branching pattern of the lower leg arteries is essential for deciding preoperatively whether the fibular artery can be harvested as part of a fibular graft. In the case of anatomic vascular variance, where the fibular artery mainly provides arterial supply to the foot, harvesting is prohibited. Without preoperative imaging, the risk of accidentally harvesting a fibular artery supplying the foot is very high. In a small proportion of our patient population, the TFT was longer than 5 cm. As the length of the TFT increases, the potential fibula pedicle that can be transplanted shortens. This can make fibular grafting more difficult [33].

In agreement with a recent meta-analysis [34] investigating stenoses in patients with peripheral arterial disease, we were able to reliably detect stenoses in our patient population by QISS-MRA. Due to the excellent visualization of the lower leg arteries, which was technically confirmed by a high SNR and CNR, stenoses could be reliably assessed along the entire vessel length. It is just as important to assess stenosis of the lower leg arteries as it is to evaluate the anatomical branching pattern for preoperative planning of the fibular graft. For example, harvesting the fibular artery is not recommended if the other arteries supplying the foot have hemodynamically relevant stenoses. A major diagnostic advantage for assessing stenosis with QISS-MRA is that it is possible to display the vessels via a 3D reconstruction in transverse as well as in coronary and sagittal spatial directions. Unlike ultrasound, it is also possible to view the entire length of the vessel at once, making it possible to visually detect multiple stenoses faster.

For osteoseptocutaneous fibular grafts, the presence of the fibular perforators is crucial. [6,9]. Therefore, it is of particular importance that the harvested graft includes at least one strong perforator vessel to maintain the perfusion and vitality of the skin island. Normally, oral and maxillofacial surgeons examine the surgical area on the lower leg just before making the skin incision directly in the operating room using ultrasound. In this way, strong fibular perforators are selected (Figure 7).

However, ultrasound examinations have the disadvantage of being highly subjective and operator-dependent [28]. For this reason, the preoperative assessment of fibular perforators using QISS-MRA provides important information for surgeons as to which area of the lower leg they can expect to find strong perforators. This can significantly shorten the duration of surgery and improve the surgical outcome with a vital graft. Several previous studies by different research groups demonstrated that CE-CTA and CE-MRA are able to detect fibular perforators [6,9,12,35,36]. With the exception of one study [36], the preoperatively determined number and localization of fibular perforators agreed with the intraoperative results. This study demonstrates that QISS-MRA can be performed to visualize fibular perforators without the use of contrast agents. Both the number and location of fibular perforators can be assessed with QISS-MRA. The high spatial resolution of 0.5 × 0.5 × 3 mm^3^ and excellent arterial contrast of QISS-MRA make it possible to detect perforator vessels as small as 1–2 mm in diameter.

We were unable to detect fibular perforators in a total of 11 patient legs. In 4 of these 11 legs, fibular grafting was performed without subsequent graft failure, although the perforators were not visible. For this reason, we hypothesize that despite the excellent image quality, the already very high spatial resolution, and the predominant absence of artifacts of QISS-MRA, not all anatomically present fibular perforators, especially very small ones, can be detected. In the future, the use of even higher spatial resolution QISS-MRA pulse sequences, e.g., at 3T MRI, would be useful in this regard. QISS-MRA can distinguish between purely septocutaneous, purely musculocutaneous, and between fibular perforators, which mainly run within the septum and have a short course in muscle [6]. The typical course of the purely septocutaneous fibular perforators in the posterolateral intermuscular septum could be visualized by QISS-MRA (Figure 8).

Although the purely septocutaneous perforators are preferred for fibular transplantation because they are easier to harvest and do not require muscle resection, all three forms of fibular perforators are candidates for transplantation. For this reason, we did not distinguish between the three different types of perforators in our work. As shown in Figure 8, QISS-MRA can be used to follow the course of the perforators to the cutis in many cases. When this is not the case, it is probably because the perforators, like all vessels, taper in the periphery and become too small to detect.

Except for the evaluation of stair-step artifacts, interobserver agreement analysis revealed substantial to almost perfect agreement with respect to the remaining imaging artifacts, vessel grading, and stenosis scoring.

Half of our study patients (25/50) underwent fibular transplantation. Based on our radiological evaluations, all of them were suitable for this surgery. Only two of these patients developed severe complications and unfortunately eventually lost their graft. Based on our evaluations, neither a clinical nor a technical parameter could be blamed for this graft loss. The success rate of 92% agrees with a meta-analysis on the use of vascularized fibular flap in mandibular reconstruction that indicated an overall success rate of 93% [37]. The mandibular reconstruction is an extensive and time-critical surgery with a total duration of about 7–8 h. Two surgeons operate simultaneously on an anesthetized patient in the lower leg and maxillofacial area. Intraoperative ultrasonography is used to find a vessel with the strongest perforators. To correlate our imaging findings with the intraoperative findings, all variables that we collected non-invasively using non-CE QISS-MRA should be measured intraoperatively using ultrasonography. Due to the retrospective design of this study, we are unable to present the correlation between intraoperative findings and imaging results. We are aiming in a prospective follow-up study to focus on this correlation. Furthermore, we aim to develop a predictive scoring system that allows us to estimate the risk for vascular complications and flap-failure based on preoperative radiological parameters.

CT systems are more available compared to MRI systems. Nevertheless, CT imaging is associated with intrinsic disadvantages, including radiation exposure, which is associated with higher cancer risks, and the need for iodinated contrast agents. Therefore, inappropriate use of CE-CTA raises concerns, particularly in young and pregnant patients, as well as in patients suffering from chronic kidney disease [38]. To obtain a pure arterial angiogram with CE-CTA or CE-MRA, it is essential to consider the patient’s blood circulation time. In contrast, the QISS-MRA pulse sequence automatically adjusts the acquisition time to the patient’s blood circulation through ECG triggering. QISS-MRA can also be used in patients with cardiac arrhythmia because the quiescent-interval time between the 90 degree in-plane saturation pulse (applied immediately after the R wave) and the readout (and T_1_ recovery and inflow) is always the same. In contrast to CE-CTA or CE-MRA, QISS-MRA is a noninvasive method and can therefore be repeated as often as desired. The measurement time is about 1 s per slice, depending on the cardiac cycle of the examined patients. The total examination time was approximately 8 min, depending on the patient’s heart rate [39]. While the measurement time of CE-MRA is reported on paper to be approximately 60 s, the total time required to perform CE-MRA is actually longer than the time required for ECG-triggered QISS-MRA. This is because additional time is required to prepare patients for contrast injection, acquire a precontrast dataset that serves as a mask for CE-MRA, and post-process the CE dataset. In addition, QISS-MRA does not require overview imaging or timing scans and does not require patient-specific image acquisition parameters to be set [40]. Neither invasive blood sampling to determine renal function nor medical monitoring of patients by a physician during the examination is required. By not using a contrast agent, QISS-MRA can also be performed in renally impaired, pregnant, and breastfeeding patients or patients with contrast agent allergies. The omission of GBCAs eliminates the risk of nephrogenic systemic fibrosis and avoids gadolinium deposition throughout the body [13,14], for which the consequences are still unclear.

The limitation of our single-center study was the relatively small number of patients.

## 5. Conclusions

In conclusion, the anatomical branching patterns, stenoses, and fibular perforators of the lower leg arteries can be visualized using QISS-MRA for preoperatively evaluating patients for planned fibular grafts without administering GBCAs. Therefore, QISS-MRA has the potential to be used as a preoperative MRA technique for diagnosing lower extremity vascular disease.

## Figures and Tables

**Figure 1 jcm-12-01634-f001:**
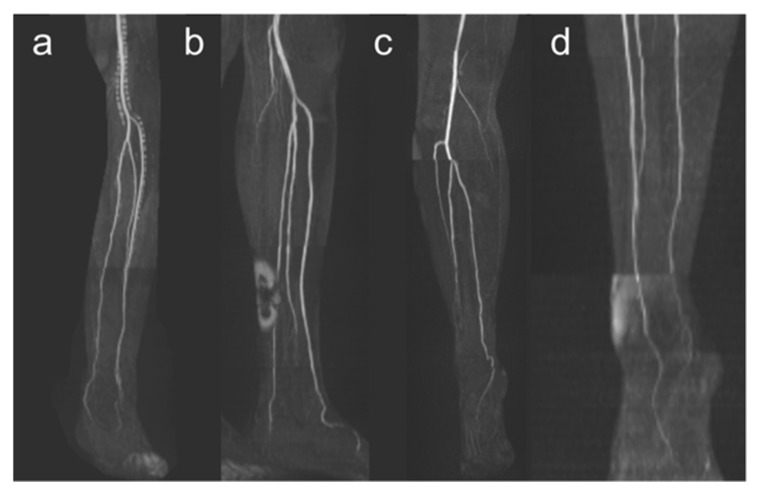
Imaging artifacts. Venous contamination is shown in panel (**a**), signal dropouts due to endoprostheses or other metallic implants in panel (**b**), stair-step artifacts in panel (**c**), and different signal intensities within a measurement slab in panel (**d**). Motion artifacts are not demonstrated here because they can only be traced in the course of several transverse slices.

**Figure 2 jcm-12-01634-f002:**
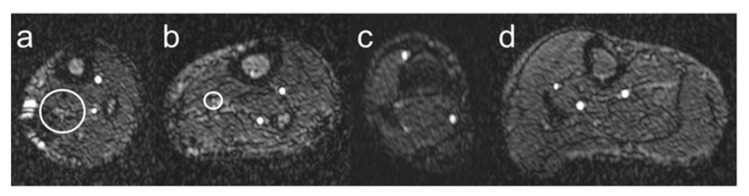
Vessel visualization. Schematic illustration of the visualization of lower leg arteries based on the grading scoring system introduced in Section 2.3. Grade 1 is shown in panel (**a**), grade 2 in panel (**b**), grade 3 in panel (**c**), and grade 4 in panel (**d**). The white circles indicate lower leg arteries with barely visible lumen (grade 1) and poorly defined vessel boundaries (grade 2).

**Figure 3 jcm-12-01634-f003:**
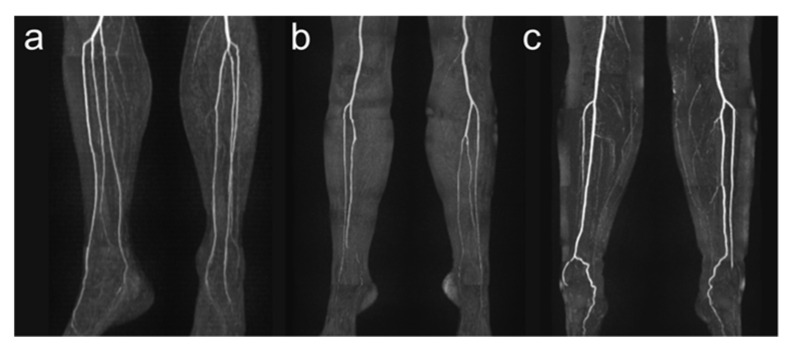
Exemplary illustration of anatomical vessel variations of lower leg arteries. Panel (**a**) shows a type I-C according to Kim et al. [19] with the first exit of the PT below the knee joint with the subsequent exit of FA and AT via a common trunk. Panel (**b**) shows a hypoplastic PT (Type III-A) on the right side, and panel (**c**) shows a hypoplastic PT on both sides. Note the supply to the foot in these cases from the FA.

**Figure 4 jcm-12-01634-f004:**
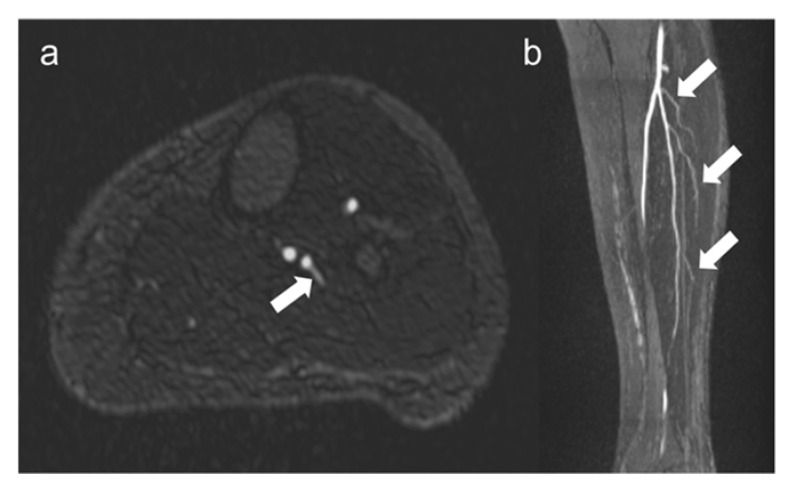
Visualization of fibular perforators using QISS-MRA. In panel (**a**), the exit of a fibular perforator is seen in transverse view (white arrow). In panel (**b**), three fibular perforators can be seen in sagittal view (white arrows). The images belong to two different patients.

**Figure 5 jcm-12-01634-f005:**
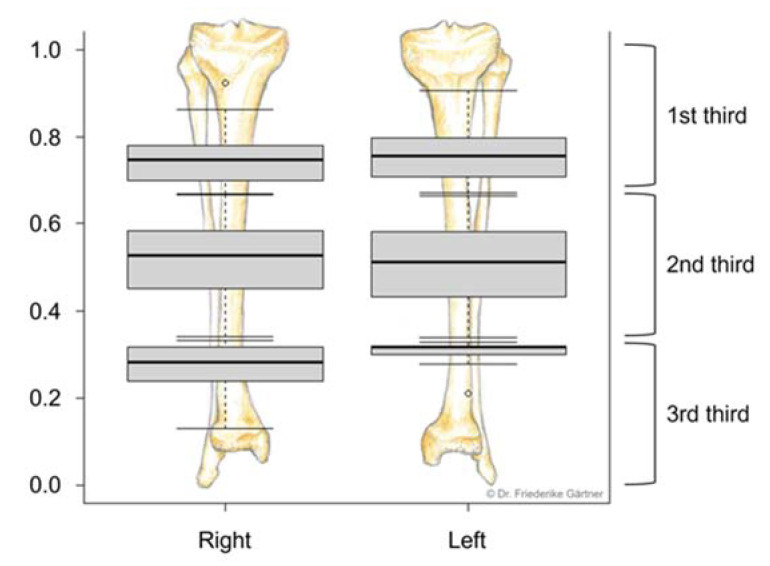
Normalized location of perforators in relation to the fibular length of an individual patient. Boxplots show the distribution of perforators relative to the 1st, 2nd, and 3rd third of the right and left fibula. This graph contains data from both radiologists. The small circle in the upper third of the right fibula and in the lower third of the left fibula indicates outlier values.

**Figure 6 jcm-12-01634-f006:**
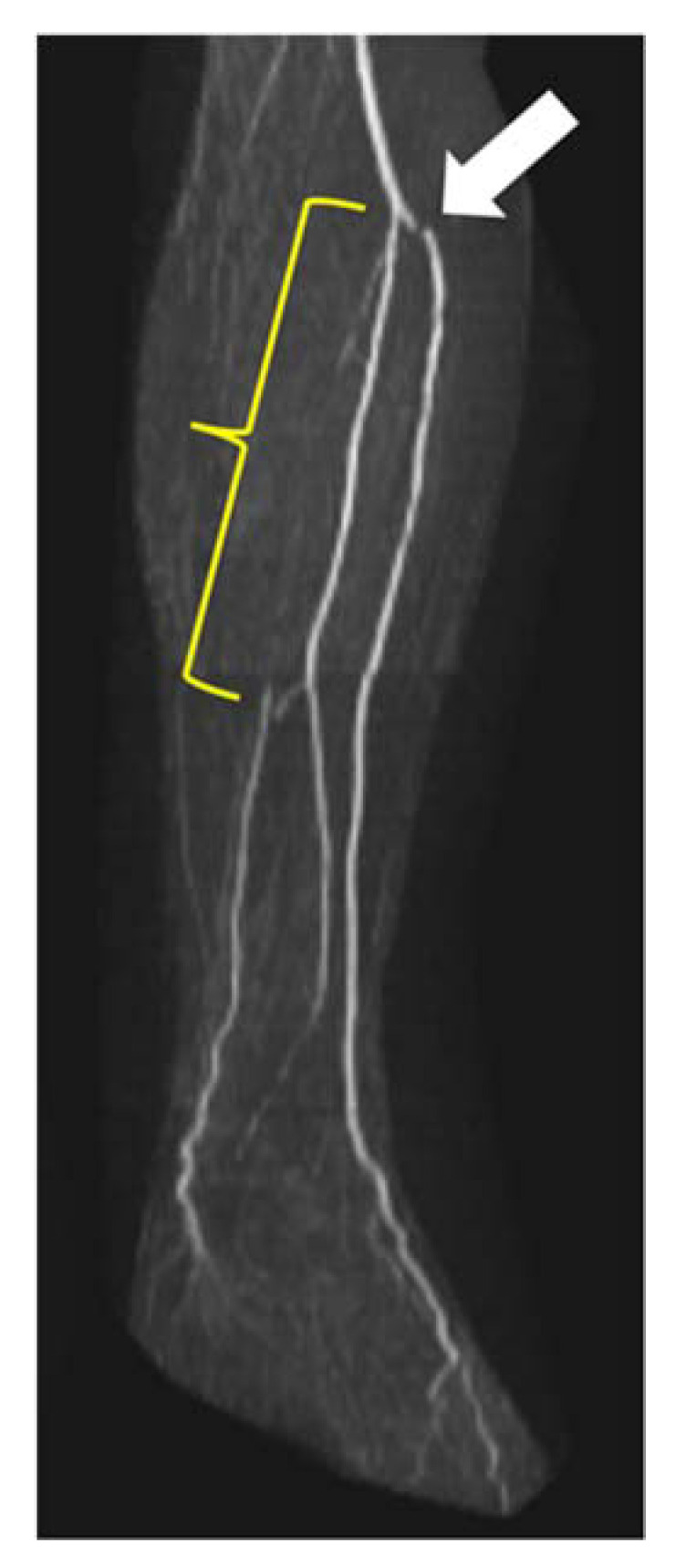
Stenosis in lower leg arteries visualized by QISS-MRA. Long-distance occlusion of the left PT (yellow curved bracket). Note the distal PT, which is refilled by the FA. Harvesting of the fibular artery is prohibited in this case because it is now the main supply to the foot. The white arrow points to a stair-step artifact.

**Figure 7 jcm-12-01634-f007:**
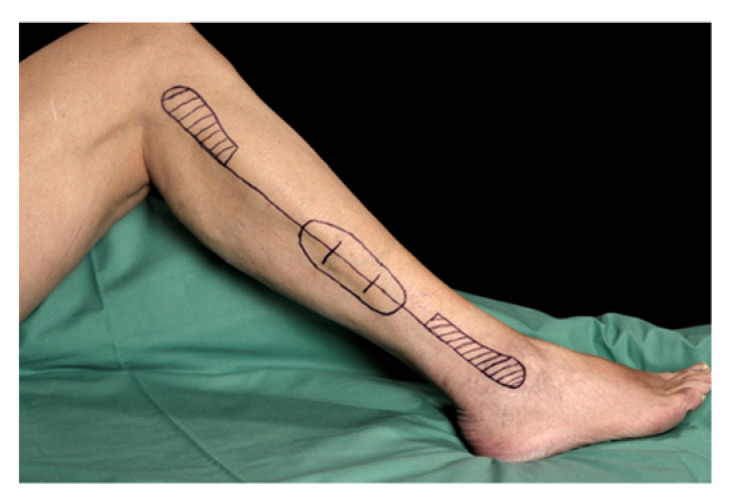
Preoperative marking of the donor site on the right leg. A safety distance of at least 8 cm should be maintained both to the fibular head and to the lateral malleolus (marked as dashed areas). The longitudinal line marks the posterior edge of the fibular bone, the origin of the posterior intermuscular septum that contains the septocutaneous perforators. Based on the preoperative imaging, the perforators are marked (typical location in the third quarter of the fibula). The skin island is outlined dependent on the location of the perforators such that at least one strong perforator is certainly included.

**Figure 8 jcm-12-01634-f008:**
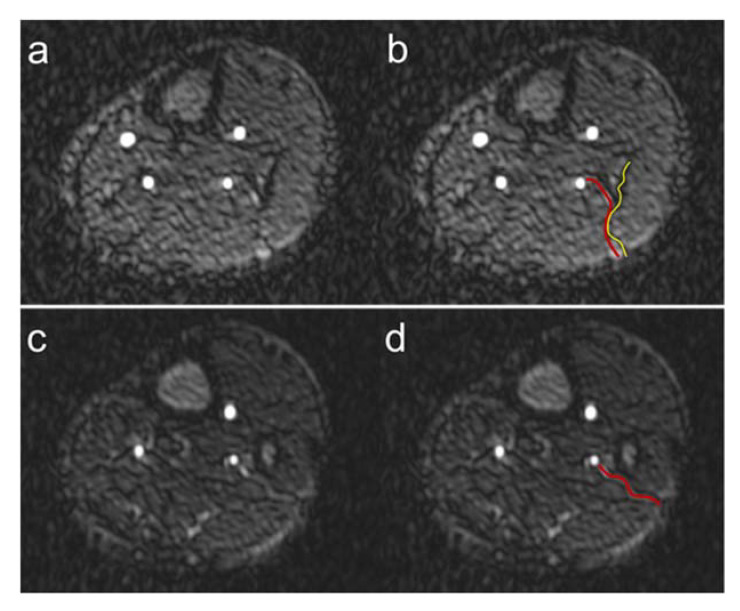
Course of fibular perforators. Exemplary course of a septocutaneous fibular perforator in the posterolateral intermuscular septum (**a**) in a 62-year-old male patient. The red line in panel (**b**) shows the course of the perforator, the yellow line the posterolateral intermuscular septum. Exemplary course of a septocutaneous fibular perforator to the cutis (**c**) in a 49-year-old male patient. The red line in panel (**d**) shows the course of the perforator to the cutaneous level.

**Table 1 jcm-12-01634-t001:** Demographic and Clinical Characteristics Data of the Study Population. The values are given either as a number (n) and percentage or as the median with interquartile range (IQR).

Variable	Value
**Demographic data**	
Female gender, n (%)	21 (42)
Median Age [IQR] (years)	65.8 [58.8, 69.4]
Median Weight [IQR] (kg)	75.0 [63.5, 83.0]
Median BMI [IQR] (kg/m^2^)	24.3 [22.1, 27.9]
**Clinical data**	
Smoker, n (%)	23 (46)
Diabetes (Type 2), n (%)	6 (12)
Hypercholesterolemia, n (%)	6 (12)
Arterial hypertension, n (%)	23 (46)
Obesity, n (%)	19 (38)
Overweight (BMI: 25–29.9 kg/m^2^)	10
Obesity grade I (BMI: 30–34.9 kg/m^2^)	5
Obesity grade II (BMI: 35–39.9 kg/m^2^)	3
Obesity grade III (BMI: ≥40 kg/m^2^)	1
Renal insufficiency, n (%)	0 (0)
Median GFR value [IQR] (mL/min/1.73 m^2^)	85.5 [70.8, 94.8]
**Underlying tumor diagnosis**	
Oral squamous cell carcinoma, n (%)	37 (74)
Osteoradionecrosis, n (%)	3 (6)
Osteochemonecrosis, n (%)	2 (4)
Keratocystic odontogenic tumor, n (%)	2 (4)
Ameloblastoma, n (%)	2 (4)
Osteoradiochemonecrosis, n (%)	1 (2)
Adenoid cystic carcinoma, n (%)	1 (2)
Osteomyelitis, n (%)	1 (2)
Mucoepidermoid carcinoma mandible, n (%)	1 (2)

**Table 2 jcm-12-01634-t002:** Signal-to-noise ratio (SNR) and contrast-to-noise ratio (CNR) determined in different vessels. The values are given as the median with the interquartile range (IQR); *p*-values were determined using the Wilcoxon signed-rank test.

Vessels	Right	Left	*p*-Value
**SNR**			
Aortic bifurcation	192.4 [137.5, 248.9]		
Popliteal artery	259.7 [195.9, 352.6]	276.6 [201.5, 368.3]	0.74
Anterior tibial artery	328.6 [246.6, 493.7]	289.4 [230.2, 438.2]	0.18
Fibular artery	339.7 [222.7, 455.9]	324.0 [219.8, 431.1]	0.53
Posterior tibial artery	337.7 [259.1, 486.2]	307.9 [219.9, 442.8]	0.26
**CNR**			
Anterior tibial artery	243.2 [183.7, 383.8]	248.3 [187.2, 368.7]	0.77
Fibular artery	265.2 [176.9, 356.8]	280.1 [184.5, 367.0]	0.66
Posterior tibial artery	268.4 [185.9, 360.6]	262.0 [191.8, 386.1]	0.91

**Table 3 jcm-12-01634-t003:** Number of perforators and grading of image quality of fibular artery (FA), anterior tibial artery (AT), and posterior tibial artery (PT) determined in lower right and left legs by two observers. The values are given as the median with the interquartile range (IQR); *p*-values were determined using the Wilcoxon signed-rank test.

Variable	Right Leg	Left Leg	*p*-Value (Right vs. Left)
**Perforators, n**			
Observer 1	2 [1, 3]	3 [1, 3]	0.36
Observer 2	2 [1, 3]	3 [1, 3]	0.46
Interobserver *p*-value	0.79	0.97	
Both observers	2 [1, 3]	3 [1, 3]	0.24
**Grading of vessels**			
**FA Grading**			
Observer 1	4 [4, 4]	4 [4, 4]	0.97
Observer 2	4 [4, 4]	4 [4, 4]	0.26
Interobserver *p*-value	0.61	0.57	
Both observers	4 [4, 4]	4 [4, 4]	0.41
**AT Grading**			
Observer 1	4 [4, 4]	4 [4, 4]	0.79
Observer 2	4 [4, 4]	4 [4, 4]	0.87
Interobserver *p*-value	0.83	0.75	
Both observers	4 [4, 4]	4 [4, 4]	0.75
**PT Grading**			
Observer 1	4 [4, 4]	4 [4, 4]	0.54
Observer 2	4 [4, 4]	4 [4, 4]	0.70
Interobserver *p*-value	0.63	0.80	
Both observers	4 [4, 4]	4 [4, 4]	0.50

**Table 4 jcm-12-01634-t004:** Evaluation of imaging artifacts and the presence or absence of stenosis in the fibular artery (FA), anterior tibial artery (AT), and posterior tibial artery (PT) in the lower right and left legs by two observers; *p*-values were determined using the Wilcoxon signed-rank test.

Variable	Right Side			Left Side		
	Observer 1	Observer 2	*p*-Value	Observer 1	Observer 2	*p*-Value
	%			%		
**Imaging artifacts**						
a	2.0 (1/50)	2.0 (1/50)	0.82	2.0 (1/50)	4.0 (2/50)	0.29
b	2.0 (1/50)	2.0 (1/50)		2.0 (1/50)	2.0 (1/50)	
c	70.0 (35/50)	76.0 (38/50)		64.0 (32/50)	76.0 (38/50)	
d	10.0 (5/50)	6.0 (3/50)		8.0 (4/50)	6.0 (3/50)	
e	8.0 (4/50)	10.0 (5/50)		4.0 (2/50)	10.0 (5/50)	
**Scoring scale**						
**FA**						
0	96.0 (48/50 vessels)	94.0 (47/50 vessels)	0.08	100.0 (50/50 vessels)	94.0 (47/50 vessels)	0.56
1						
A	2.0 (1/50 vessels)	2.0 (1/50 vessels)		0	4.0 (2/50 vessels)	
B	0	0		0	2.0 (1/50 vessels)	
C	0	2.0 (1/50 vessels)		0	0	
AB	0	0		0	0	
ABC	0	0		0	0	
Missing vessels	2.0 (1/50 vessels)	2.0 (1/50 vessels)		0	0	
**AT**						
0	94.0 (47/50 vessels)	98.0 (49/50 vessels)	0.30	94.0 (47/50 vessels)	96.0 (48/50 vessels)	0.16
1						
A	0	2.0 (1/50 vessels)			0	
B	0	0		2.0 (1/50 vessels)	0	
C	4.0 (2/50 vessels)	0		2.0 (1/50 vessels)	0	
AB	0	0		0	0	
ABC	2.0 (1/50 vessels)	0		0	0	
Missing vessels		0		2.0 (1/50 vessels)	4.0 (2/50 vessels)	
**PT**						
0	96.0 (48/50 vessels)	90.0 (45/50 vessels)	0.03	92.0 (46/50 vessels)	92.0 (46/50 vessels)	1
1						
A	0	2.0 (1/50 vessels)		0	0	
B	0	2.0 (1/50 vessels)		0	0	
C	0	4.0 (2/50 vessels)		0	0	
AB	0	0		2.0 (1/50 vessels)	2.0 (1/50 vessels)	
ABC	0	2.0 (1/50 vessels)		2.0 (1/50 vessels)	2.0 (1/50 vessels)	
Missing vessels	4.0 (2/50 vessels)	0		4.0 (2/50 vessels)	4.0 (2/50 vessels)	

**Table 5 jcm-12-01634-t005:** Interobserver agreement for the technical and clinical evaluation of Cartesian QISS-MRA. Data are presented as agreement, and 95% confidence intervals are shown in parentheses. Anterior tibial artery is abbreviated to AT, posterior tibial artery to PT, and fibular artery to FA.

Variable	Interobserver Agreement
	Both sides
Imaging artifacts	
a	0.98 (0.94–1.00)
b	1.00 (1.00–1.00)
c	0.66 (0.50–0.80)
d	0.86 (0.74–0.96)
e	0.84 (0.72–0.94)
Grading of vessel visualization	
FA	0.97 (0.96–0.99)
AT	0.97 (0.96–0.99)
PT	0.97 (0.95–0.98)
Scoring of vessel stenosis	
FA	0.97 (0.92–1.00)
AT	0.88 (0.75–0.97)
PT	0.91 (0.81–0.99)

## Data Availability

The data presented in this study are available on request from the corresponding author.

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
