# Peer review of "Evaluation of Lower Leg Arteries and Fibular Perforators before Microsurgical Fibular Transfer Using Noncontrast-Enhanced Quiescent-Interval Slice-Selective (QISS) Magnetic Resonance Angiography"

_jcm, 2023, doi:10.3390/jcm12041634_

Round 1

Reviewer 1 Report

Dear authors,

Very interesting study. Few minor suggestions - 

Line 31: Grammar - please replace "preoperative" to "preoperatively"

Line 223 : Unclear phrasing "p-values were reported with a precision of 10−5".

Section 3.7: Surgical outcome - please include brief analysis regarding observations in non-CE QISS MRA relating to surgical outcome (in cases where FFF was used for surgical reconstruction

Discussion: In the final discussion section, in addition to the compelling reasons to adopt this method for planning fibular grafting, please also include potential caveats and/or confounding artefacts associated with QISS MRA.

Author Response

Reply to the Review (JCM 2136314)

Thank you for your feedback and suggestions for improving the quality of our manuscript. The following points correspond to your comments and those in the revised manuscript.

The comments and suggestions of Reviewer No. 1:

Dear authors,

Very interesting study.

Thank you very much for your important remark.

Few minor suggestions -

  1. Line 31: Grammar - please replace "preoperative" to "preoperatively"

We edited this word in the revised manuscript.

  1. Line 223 : Unclear phrasing "p-values were reported with a precision of 10−5".

The minus 5 should be the power of 10 in the manuscript, not 10 minus 5. The value 5 indicated the number of decimal places for the precession of calculated p-value (p=0.00001). We have edited the notation in the revised manuscript on page 6.

  1. Section 3.7: Surgical outcome - please include brief analysis regarding observations in non-CE QISS MRA relating to surgical outcome (in Discussion: In the final discussion section, in addition to the compelling reasons to adopt this method for planning fibular grafting, please also include potential caveats and/or confounding artefacts associated with QISS MRA.

Thank you for this comment. We have described in detail the potential imaging artefacts associated with non-CE QISS-MRA using Figure 1 in the Materials and Methods section. In the Results and Discussion section under "Technical Aspects", we have presented and discussed the impact of these artefacts on the obtained vascular results. There are no other artefacts associated with non-CE QISS-MRA that can be additionally discussed.

The purpose of this retrospective study was to investigate the feasibility and clinical value of QISS-MRA for reliable visualization of anatomical branching patterns of the lower leg arteries and fibular perforators. Postoperative outcome of fibular grafting based on assessment of vascular complications and flap failures was compared with preoperative assessment of the lower leg arteries using QISS-MRA. We focused on the radiological technique of QISS-MRA and performed a retrospective analysis of clinical outcomes. We analyzed therefore the relevant surgical outcomes in detail in the Results section under "surgical outcomes" and discussed them in the Discussion section under "surgical aspects".

Reviewer 2 Report

The Authors aimed to evaluate the feasibility and clinical value of non-contrast-enhanced (CE) Quiescent-Interval Slice-Selective (QISS)-MRA for reliably visualizing the anat omy and patency of the lower leg arteries and for preoperative determining the presence, number,  and location of fibular perforators. 

The topic is extremely interesting and the study is well designed.

I would evaluate CE enhanced CT scan of the same patients to assess any differences.

Please discuss further on advantages and disadvantages of evaluation of CT and MRI. For sure, CT is more available but MRI can be performed even in allergic patients reducing risks.

Also, I would love to see a correlation between imaging findings and intraoperative findings

Author Response

Reply to the Review (JCM 2136314)

Thank you for your feedback and suggestions for improving the quality of our manuscript. The following points correspond to your comments and those in the revised manuscript.

The comments and suggestions of Reviewer No. 2:

The Authors aimed to evaluate the feasibility and clinical value of non-contrast-enhanced (CE) Quiescent-Interval Slice-Selective (QISS)-MRA for reliably visualizing the anatomy and patency of the lower leg arteries and for preoperative determining the presence, number, and location of fibular perforators. The topic is extremely interesting, and the study is well designed.

Thank you very much for your suggestion.

  1. I would evaluate CE enhanced CT scan of the same patients to assess any differences.

We thank you for this comment. From a scientific point of view, we would be very interested to compare CE-CTA examinations with QISS-MRA of the same patients. However, from an ethical and Radiation Protection Ordinance it is not possible to perform an additional CE-CTA examination. Moreover, as we mentioned in the introduction section, oncologic patients undergo multiple preoperative and subsequent imaging examinations due to their diseases. Therefore, it is desirable to perform non-CE-MRA examinations instead of CE-CTA examinations to avoid the adverse effects of radiation exposure and iodine-containing contrast agents. In patients with oral and maxillofacial tumors, both imaging techniques (CE-CTA and non-CE QISS-MRA) can be considered if CE-CTA apart from its side effects would provide additional and relevant vascular information compared to QISS-MRA for these patients.

In the previous studies [1–3] non-CE QISS-MRA were compared with the standard clinical imaging modalities such as CE-MRA, ultrasound, invasive angiography, and CE-CTA for the visualization of leg arteries and their abnormalities (e.g. stenosis). Based on this data, non-CE QISS-MRA provides reliable clinical results. Moreover, our in-vivo results concerning the localization of the perforators of the leg are in good agreement with the results of Martin et al. and Schuderer et al. [4, 5].

  1. Please discuss further on advantages and disadvantages of evaluation of CT and MRI. For sure, CT is more available, but MRI can be performed even in allergic patients reducing risks.

Thank you very much for your comments. We agree with you that CT systems are more available compared to MRI systems. Nevertheless, CT imaging is associated with intrinsic disadvantages, including radiation exposure, which is associated with higher cancer risks, and the need for iodinated contrast agents. Therefore, inappropriate use of CE-CTA raises concerns particularly in young and pregnant patients as well as in patients suffering from chronic kidney disease [6].

We added these aspects to the discussion section on page 7.

  1. Also, I would love to see a correlation between imaging findings and intraoperative findings

Thank you very much for this comment. With this comment you have raised a highly relevant question, that we are also interested in. Unfortunately, due to the retrospective design of this study, we are unable to present this correlation. We have focused on the radiological technique of QISS-MRA and have performed a retrospective analysis of the clinical outcome.

The mandibular reconstruction is an extensive and time-critical surgery with a total duration of about 7-8 hours. Two surgeons operate simultaneously on an anesthetized patient in the lower leg and maxillofacial area. Intraoperative ultrasonography is used to find a vessel with strongest perforators. To correlate our imaging findings with the intraoperative findings, all variables that we collected non-invasively using non-CE QISS-MRA should be measured intraoperatively.

We are aiming in a prospective follow-up study to focus on the correlation between imaging findings and intraoperative findings. Furthermore, we aim to develop a predictive scoring system that allows to estimate the risk for vascular complications and flap-failure based on preoperative radiological parameters.

We added these aspects to the discussion section on page 6.

References

[1]   Carr JC. QISS MR Angiography: An Alternative to CT Angiography for Peripheral Vascular Evaluation. JACC. Cardiovascular imaging 2017;10:1125–7.

[2]   Saini A, Wallace A, Albadawi H, Naidu S, Alzubaidi S, Knuttinen MG, et al. Quiescent-Interval Single-Shot Magnetic Resonance Angiography. Diagnostics (Basel, Switzerland) 2018;8.

[3]   Varga-Szemes A, Aouad P, Schoepf UJ, Emrich T, Yacoub B, Todoran TM, et al. Comparison of 2D and 3D quiescent-interval slice-selective non-contrast MR angiography in patients with peripheral artery disease. Magma (New York, N.Y.) 2021;34:649–58.

[4]   Martin AL, Bissell MB, Al-Dhamin A, Morris SF. Computed tomographic angiography for localization of the cutaneous perforators of the leg. Plastic and reconstructive surgery 2013;131:792–800.

[5]   Schuderer JG, Meier JK, Klingelhöffer C, Gottsauner M, Reichert TE, Wendl CM, Ettl T. Magnetic resonance angiography for free fibula harvest: anatomy and perforator mapping. International journal of oral and maxillofacial surgery 2020;49:176–82.

[6]   Smith-Bindman R, Lipson J, Marcus R, Kim K-P, Mahesh M, Gould R, et al. Radiation dose associated with common computed tomography examinations and the associated lifetime attributable risk of cancer. Archives of internal medicine 2009;169:2078–86.

Round 2

Reviewer 2 Report

the Authors made great efforts in the attempt to ameliorate their paper. No other comments now.